# Articular Tissue-Mimicking Organoids Derived from Mesenchymal Stem Cells and Induced Pluripotent Stem Cells

Zhong Alan Li [1,2,†], Jiangyinzi Shang [1], Shiqi Xiang [1], Eileen N. Li [1], Haruyo Yagi [1], Kanyakorn Riewruja [1], Hang Lin [1,3,4,*] and Rocky S. Tuan [4,5,*]

1 Center for Cellular and Molecular Engineering, Department of Orthopaedic Surgery, University of Pittsburgh School of Medicine, Pittsburgh, PA 15219, USA
2 Department of Neurobiology, University of Pittsburgh School of Medicine, Pittsburgh, PA 15213, USA
3 Department of Bioengineering, Swanson School of Engineering, University of Pittsburgh, Pittsburgh, PA 15260, USA
4 McGowan Institute for Regenerative Medicine, University of Pittsburgh School of Medicine, Pittsburgh, PA 16219, USA
5 Institute for Tissue Engineering and Regenerative Medicine, The Chinese University of Hong Kong, Hong Kong SAR, China
* Correspondence: hal46@pitt.edu (H.L.); tuanr@cuhk.edu.hk (R.S.T.)
† Current address: Department of Biomedical Engineering, The Chinese University of Hong Kong, Hong Kong SAR, China.

**Abstract:** Organoids offer a promising strategy for articular tissue regeneration, joint disease modeling, and development of precision medicine. In this study, two types of human stem cells—primary mesenchymal stem cells (MSCs) and induced pluripotent stem cells (iPSCs)—were employed to engineer organoids that mimicked bone, cartilage and adipose tissue, three key tissue components in articular joints. Prior to organoidogenesis, the iPSCs were first induced into mesenchymal progenitor cells (iMPCs). After characterizing the MSCs and iMPCs, they were used to generate cell-embedded extracellular matrix (ECM) constructs, which then underwent self-aggregation and lineage-specific differentiation in different induction media. Hydroxyapatite nanorods, an osteoinductive bioceramic, were leveraged to generate bone and osteochondral organoids, which effectively enhanced mineralization. The phenotypes of the generated organoids were confirmed on the basis of gene expression profiling and histology. Our findings demonstrate the feasibility and potential of generating articular tissue-recapitulating organoids from MSCs and iPSCs.

**Keywords:** joint-mimicking organoid; mesenchymal stem cell; induced pluripotent stem cell; bone; cartilage; osteochondral complex; adipose tissue; hydroxyapatite





## 1. Introduction

Joint diseases represent a globally leading cause of pain and physical disability. For example, osteoarthritis (OA) affects 10–12% of the world's adult population and 25% of those aged 50 years and above [1,2]. Rheumatoid arthritis (RA), another debilitating joint disease, has a global prevalence of 0.46% [3]. No disease-modifying OA drugs (DMOADs) have been developed to date, and only a few disease-modifying anti-rheumatic drugs (DMARDs) are available with varying efficacy observed in different patients [4]. An incomplete understanding of disease etiologies and pathophysiology underlies the current unmet medical need for the treatment of joint disorders [5].

Joint disease models with high physiological and clinical relevance are vital to investigating disease mechanisms and developing efficacious therapeutics. Disease models established in small and large mammals, such as mice, rats, rabbits, and horses, have long been considered clinically relevant preclinical models and play indispensable roles in both basic and translational research into joint disorders [6,7]. Over the past decade, increasing attention has been paid to more advanced in vitro disease models because of

their advantages over in vivo models [8]. Generally, in vitro models possess higher reproducibility, allow the use of human (and patient-specific, if necessary) cells, and eliminate costly and time-consuming maintenance of animals. In addition, it is more convenient and economical to use in vitro models to dissect the contribution of individual tissues to disease onset and progression [9], which is highly valuable for the identification of possible therapeutic targets. In the context of joint disorders, the degeneration of articular cartilage is often a prominent feature; in addition, subchondral bone [10,11] and intra-articular adipose tissue [12,13] are also actively involved in the pathogenesis and development of joint disorders.

Organoids have emerged as a promising biotechnology to study organ development and pathophysiology. Organoidogenesis is generally initiated based on developmental biology principles and typically involve self-organizing stem cells. Organoids have been produced to mimic a wide range of human organs, including brain [14], lung [15], kidney [16], liver [17], pancreas [18], intestine [19], prostate [20], etc. However, the development of musculoskeletal organoids, such as those mimicking articular tissues, is still in relative infancy [21–24]. Stem cells commonly used to derive organoids include adult tissue derived mesenchymal stem cells (MSCs) and induced pluripotent stem cells (iPSCs) [23,25]. MSCs are multipotent cells present in various tissues of the adult body. However, these cells have limited self-renewal capability and experience aging and senescence at higher passages [26]. In contrast, iPSCs are produced by reprogramming adult somatic cells and may be expanded indefinitely, thus representing a virtually unlimited supply of cells, obviating the need of harvesting cells from multiple tissues or donors. Thus, iPSCs hold great potential in personalized disease modeling and treatment.

We previously employed a two-step process recapitulating endochondral ossification to engineer bone organoids from human bone marrow-derived MSCs embedded in their own secreted extracellular matrix (ECM) [23]. In the current study, a different approach, which mimics intramembranous ossification [27], was taken to directly differentiate MSCs into osteogenic progenitors and induce mineralization in the cell-ECM constructs by leveraging osteoinductive bioceramic nanoparticles. The engineering of three-dimensional (3D) MSC-ECM-derived cartilaginous and adipose organoids was also demonstrated. Furthermore, iPSC-derived, multipotent mesenchymal progenitor cells (iMPCs) were utilized in parallel to obtain organoids mimicking these three articular tissues as well as the osteochondral complex. The MSC- and iPSC-derived organoids displayed phenotypic characteristics of tissues of the articular joint and can be potentially used, in either separate or coupled forms, as in vitro models to investigate joint disease mechanisms and evaluate drug safety and efficacy.

## 2. Materials and Methods

### 2.1. Isolation and Culture of MSCs

The human bone marrow-derived MSCs used in this study were isolated from the surgical waste (femoral head and trabecular bone) of de-identified patients undergoing total hip arthroplasty (THA) with Institutional Review Board (IRB) approval (University of Washington and University of Pittsburgh), as described in our previous study [28]. In total, 20 donors were recruited, and the harvested cells were pooled [9]. The MSCs were expanded in growth medium (GM-MSC, composition provided in Table S1), dissociated by treatment with trypsin-EDTA (Thermo Fisher, Waltham, MA, USA) when reaching 70–80% confluence, and used at passage 5 (P5).

### 2.2. Differentiation of iPSCs into iMPCs

Human iPSCs were obtained by reprograming Passage 3 bone marrow MSCs via lentiviral overexpression of *Sox2*, *Oct3/4*, *c-Myc*, and *Klf4* [29]. A protocol previously established in our lab was used to induce iPSC differentiation into iMPCs [29–31]. Briefly, STEMdiff-ACF Mesenchymal Induction Medium (STEMCELL Technologies, Vancouver, BC, Canada) was used to culture the iPSCs for three days. The medium was then re-

placed with MesenCult-ACF Plus Medium (STEMCELL Technologies). After another three days, the cells were detached, seeded onto flasks coated with Animal Component-Free Cell Attachment Substrate (STEMCELL Technologies), and expanded in MesenCult-ACF Plus medium. Upon reaching ~80% confluence, the iMPCs were dissociated with ACF Enzymatic Dissociation Solution and ACF Enzyme Inhibition Solution (STEMCELL Technologies). Subsequently, the iMPCs were maintained in regular tissue culture flasks with growth medium (GM-iMPC, composition provided in Table S1). iMPCs at P4 or P5 were used in this study.

### 2.3. Characterization of MSCs and iMPCs

To evaluate the colony-forming ability of the cells, 100 MSCs or iMPCs were seeded in a 100 mm Petri dish. After 14 days of culture, the cells were stained with 0.5% Crystal Violet (Sigma-Aldrich, St. Louis, MO, USA) in methanol (Thermo Fisher) for 8 min. Purple colonies were manually counted after removing the excess dye by repeated rinsing with phosphate-buffered saline (PBS, Gibco).

The surface epitopes on the cells were examined by flow cytometry (BD FACS Aria™ II cell sorter; BD Biosciences, Franklin Lakes, NJ, USA). The cells were pre-incubated in PBS containing 2% fetal bovine serum (FBS) and FITC-conjugated mouse anti-human antibodies against cluster of differentiation 31 (CD31), CD34, CD45, CD73, and CD90 (BD Pharmingen™, San Jose, CA, USA) for 30 min.

The multipotency of the MSCs and iMPCs was evaluated by the tri-lineage differentiation assay. The cells were plated in six-well plates at 20,000 cells/cm$^2$ and cultured in osteo-, chondro- and adipo-induction media (abbreviated as OIM, CIM, and AIM, respectively; medium compositions are provided in Table S1). After 21 days, the cells cultured in OIM, CIM, and AIM were fixed in 4% paraformaldehyde (ThermoFisher) and stained with Alizarin Red (Rowley Biochemical, Danvers, MA, USA), Alcian Blue (EKI, Joliet, IL, USA), and Oil Red O (Sigma-Aldrich, St. Louis, MO, USA) dyes, respectively. For comparison purposes, MSCs and iMPCs were also cultured in control media (Table S1).

### 2.4. Growing MSC- and iMPC-Based Organoids

Figure 1 presents a schematic of the process of engineering different types of organoids. MSCs and iPSCs were first seeded in six-well tissue culture plates (Corning, Glendale, AZ, USA) at a density of ~0.2 × 10$^6$ and ~0.4 × 10$^6$ cells/well, respectively. Once the cells cultured in the corresponding GM were fully confluent, the medium was supplemented with 50 μg/mL L-ascorbic acid (AA; Sigma-Aldrich). After another ten days, copious ECM was deposited, forming a cell-embedded ECM layer. To obtain bone organoids (but not cartilage or adipose organoids), hydroxyapatite nanorods (HANRs), a highly osteoinductive nanosized bioceramic, were added at 0.9 μg/well on the last day to promote osteogenesis. The HANRs were synthesized from calcium hydroxide and orthophosphoric acid (both from Merck, Darmstadt, Germany) based on the following reaction: $10\ Ca(OH)_2 + 6\ H_3PO_4 \rightarrow Ca_{10}(PO_4)_6(OH)_2 + 18\ H_2O$. HANRs were obtained by evaporating the water and grinding the dried cake with a mortar and pestle [32].

At the end of the ten days of AA treatment, the cells were trypsinized (Invitrogen, Carlsbad, CA, USA) for 5–7 min, which caused the contraction and detachment of the ECM. Each of the loose cell-ECM constructs was then rinsed twice with the corresponding GM, transferred with a pipette to one well in low-binding 96-well conical plate (Corning), and centrifuged at 300× *g* for one minute. After another day of culture in AA-supplemented GM, the medium was changed to experimentally designated media to induce organoidogenesis over four weeks. The media utilized to generate MSC-derived bone, cartilage, and adipose organoids are denoted as OM-MSC, CM-MSC, and AM-MSC, respectively; the corresponding media used for iMPCs were named OM-iMPC, CM-iMPC, and AM-iMPC, respectively. The compositions of the induction media as well as control media are provided in Table S2.

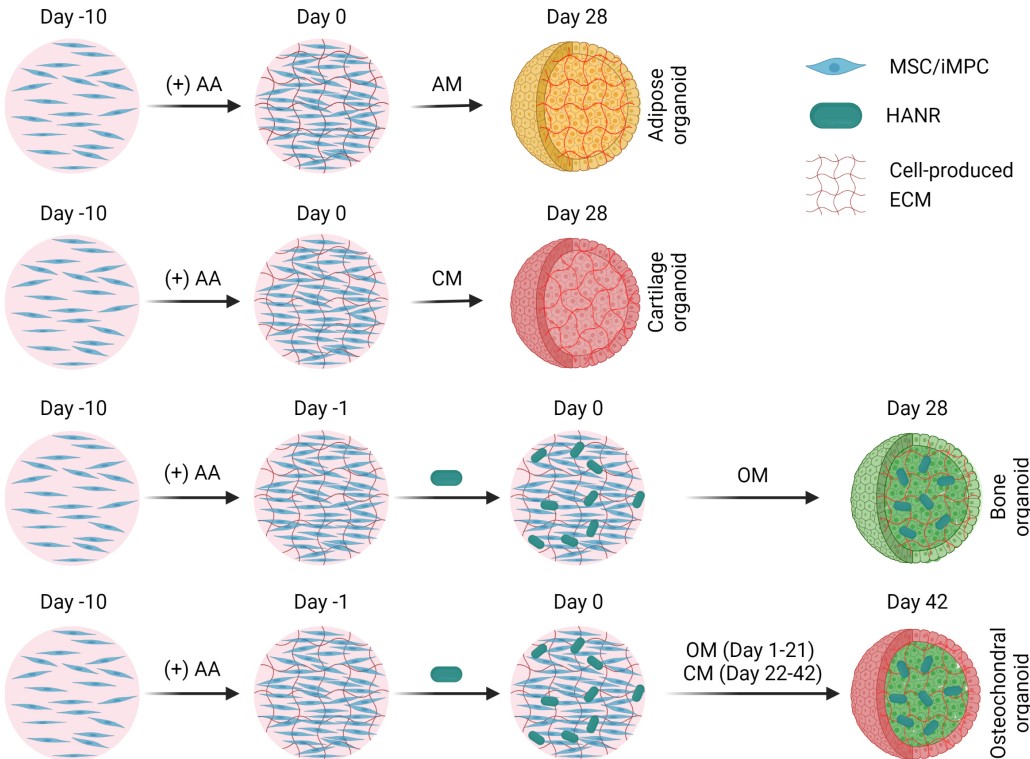

**Figure 1.** Schematic depicting the process of producing different types of organoids from MSCs or iMPCs. AM, adipogenic medium. CM, chondrogenic medium. OM osteogenic medium. HANR, hydroxyapatite nanorod. ECM, extracellular matrix.

To obtain osteochondral organoids, HANR-containing iMPC-ECM constructs were first cultured in OM-iMPC for 21 days to promote osteogenesis. Subsequently, the culture medium was switched to CM-iMPC and used for the next 21 days to induce cartilage formation on the surface.

### 2.5. Real-Time Quantitative Polymerase Chain Reaction (RT-qPCR)

An RNeasy Plus Universal Kit (Qiagen, Hilden, Germany) was used for RNA extraction. The SuperScript™ IV VILO™ Master Mix (Invitrogen, Waltham, MA, USA) was then used for reverse transcription to obtain complementary DNA (cDNA). RT-qPCR was carried out using SYBR green chemistry (Applied Biosystems, Waltham, MA, USA) on a QuantStudio 3 RT-qPCR system (Applied Biosystems). The primer sequences (IDT, Newark, NJ, USA) used for RT-qPCR are provided in Supplementary Table S3.

### 2.6. Histology

The bone, cartilage, and osteochondral organoids were formalin-fixed and paraffin-embedded, sectioned to 6 µm thick slices, and stained with Alizarin Red and Safranin O/Fast Green (Sigma-Aldrich). The adipose organoids were fixed in 4% paraformaldehyde, cryo-embedded in Cryo-Gel (Leica, Wetzlar, Germany), sectioned at a thickness of 10 µm, and stained with Oil Red O or BODIPY™ 493/503 (Invitrogen). The stained sections were mounted and imaged on a Nikon Eclipse E800 upright microscope (Nikon, Tokyo, Japan).

### 2.7. Immunohistochemistry

Immunohistochemistry (IHC) was carried out to comparatively analyze the presence of COL2 and OCN in the iMPC-derived osteochondral organoids and native osteochondral tissues. To obtain the native osteochondral tissue sections, osteochondral plugs were harvested from the surgical waste of a 56-year-old male donor who underwent THA. The

osteochondral plugs were fixed in 4% paraformaldehyde, cryo-embedded in Cryo-Gel, and sectioned at a thickness of 12 μm.

Antigen retrieval was carried out for the osteochondral organoid sections following deparaffinization and rehydration. To detect COL2, the samples were incubated in a combined hyaluronidase and chondroitinase (both purchased from Sigma-Aldrich) solution at 37 °C for 30 min. For OCN detection, the sections were treated with 90 °C sodium citrate solution (eBioscience, San Diego, CA, USA) for 20 min. After antigen retrieval, the organoid sections, together with thawed cryosections of the native osteochondral tissues, were incubated in primary antibodies targeting COL2 (1:150 dilution; MA5-12789, Invitrogen) or OCN (1:50 dilution; MAB1419, R&D Systems, Minneapolis, MN, USA) overnight. Mouse immunoglobulin G (IgG) isotypes (0.5 μg/mL) were used in place of the primary antibodies as negative controls. A biotinylated anti-mouse/rabbit IgG secondary antibody was used, and a Vector NovaRED substrate kit (both from Vector Laboratories, Burlingame, CA, USA) was employed for signal visualization. The sections were counter-stained with Hematoxylin QS (Vector Laboratories). Sample imaging was carried out on a Nikon Eclipse E800 upright microscope.

### 2.8. Statistical Analysis

At least three replicates were used for all experiments. Data analysis was conducted with Prism 9 (GraphPad, San Diego, CA, USA), with the specific data analysis method specified in each figure legend. Data are presented as mean ± standard deviation, with statistical difference denoted by * ($p < 0.05$), ** ($p < 0.01$), *** ($p < 0.001$), and **** ($p < 0.0001$).

## 3. Results

### 3.1. Properties of MSCs and iMPCs

Flow cytometry results showed that the majority of MSCs (99.9% and 99.6%, respectively) and iMPCs (96.2% and 100%, respectively) were positive for CD73 and CD90 (Figure 2A). In addition, all MSCs and iMPCs were negative for CD31, CD34, and CD45 (Figure 2A). Both MSCs and iMPCs possessed colony-forming ability (Figure 2B). Furthermore, trilineage differentiation assay results supported the multipotency of both types of cells (Figure 2C). These results confirmed that both cell sources displayed key characteristics of MSCs [33].

### 3.2. MSC-Derived Organoids

3.2.1. MSC-Derived Bone Organoids

The osteo-induction ability of OM-MSC (Table S2), a medium commonly used to differentiate MSCs into osteoblastic cells, was first analyzed and compared with that of HANRs (Figure 3A). Interestingly, RT-qPCR results showed that compared with those cultured in OM-MSC (without HANR addition), MSCs cultured in the presence of HANRs in GM expressed higher levels of key osteogenic marker genes, including ostepontin (*OPN*), bone sialoprotein 2 (*BSP2*), bone morphogenetic protein 2 (*BMP2*), and RUNX family transcription factor 2 (*RUNX2*) (Figure 3B).

Next, HANR-encapsulated MSC-ECM constructs were cultured in OM-MSC (schematic of preparation process shown in Figure 1) or GM to comparatively evaluate the quality of osteogenesis. It was found that compared with the GM group, the constructs grown in OM-MSC showed upregulated expression of osteocalcin (*OCN*), alkaline phosphatase (*ALP*), and *RUNX2* (Figure 4A). Collectively, these results suggest that HANRs and OM-MSC synergistically promoted MSC osteogenesis.

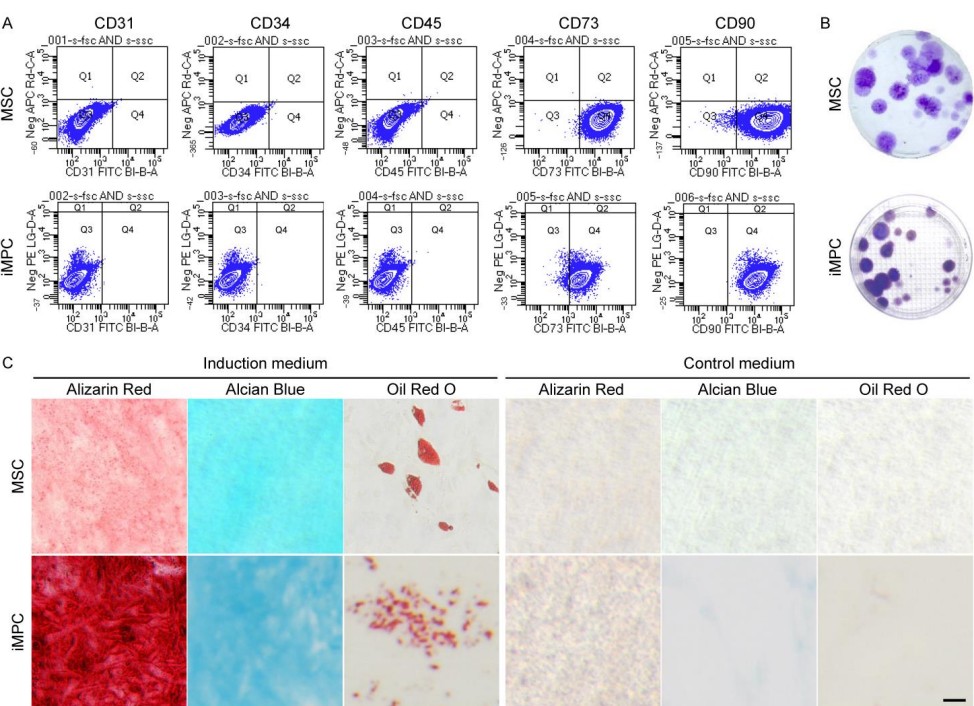

**Figure 2.** Characterization of MSCs and iMPCs. (**A**) Flow cytometry analysis of the surface epitope profiles of MSCs and iMPCs. (**B**) Representative images of the colonies formed from 100 initially seeded cells in the colony-forming unit assay. The average number of colonies was 25 for MSCs and 21 for iMPCs. (**C**) Histological images from trilineage differentiation assay. Scale bar = 50 μm.

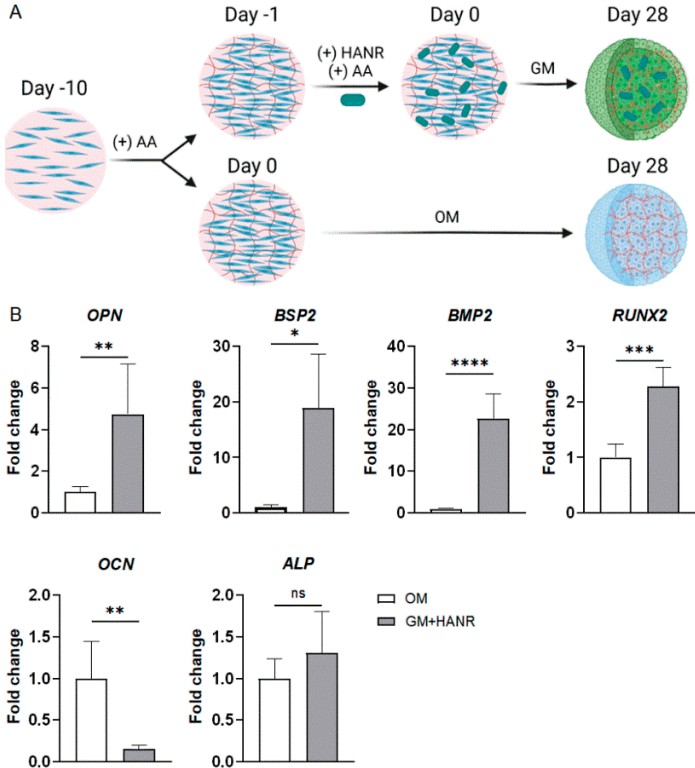

**Figure 3.** Osteogenic induction via HANR addition is more effective to produce bone organoids from MSCs than the common practice of culture in OM. (**A**) Schematic of experimental design. (**B**) Expression of osteogenic gene markers in the two groups of samples. Statistical analysis was conducted using Student's *t*-test. N ≥ 3 replicates. *, $p < 0.05$; **, $p < 0.01$; ***, $p < 0.001$; ****, $p < 0.0001$.

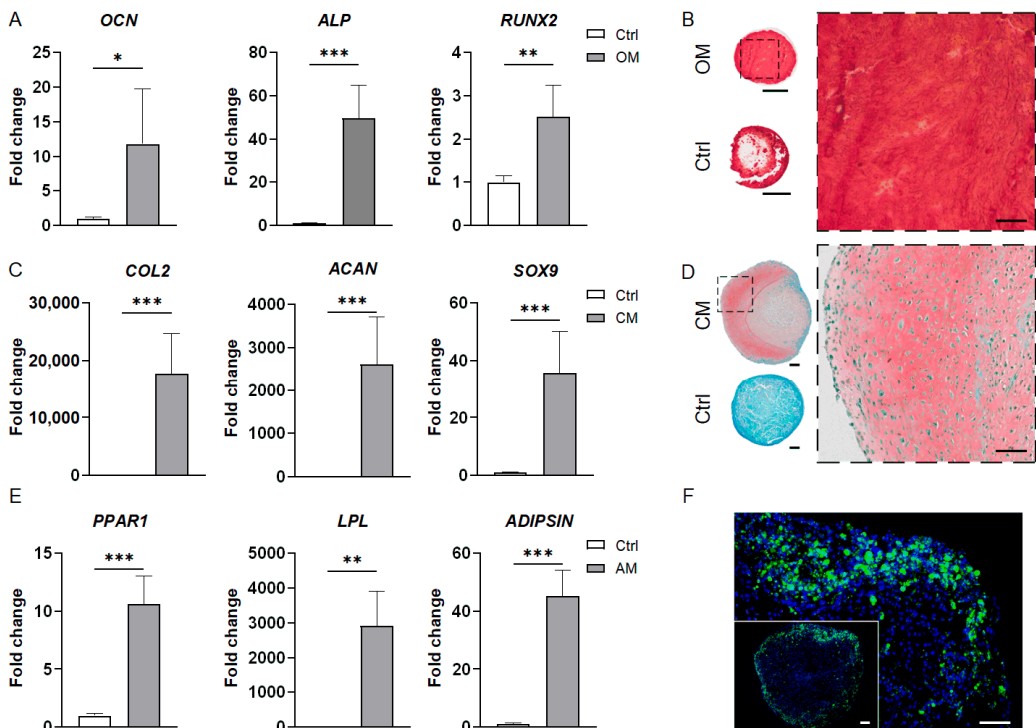

**Figure 4.** Characterization of MSC-derived bone, cartilage, and adipose organoids. (**A**) RT-qPCR results showing the expression of osteogenic markers in the bone organoids. Data are normalized to the control group. (**B**) Alizarin Red staining images of the bone organoid and control sample. The image on the right is a magnified view of the dashed line square in the whole-section image on the left. (**C**) Expression levels of chondrogenic marker genes in the cartilage organoids as compared to the control group (set at 1). (**D**) Safranin O/Fast Green staining images of the cartilage organoid and control group. The image on the right is a magnified view of the dashed line square in the whole-section image on the left. (**E**) Expression levels of adipogenic marker genes in the adipose organoids compared to the control group (set at 1). (**F**) BODIPY staining images of the adipose organoid. All RT-qPCR data were analyzed by Student's *t*-test (N = 4 replicates). *, $p < 0.05$; **, $p < 0.01$; ***, $p < 0.001$. Scale bar = 500 µm for the wholemount section images and 100 µm for others.

Alizarin Red staining was carried out to visualize calcium deposits in the bone organoids. Figure 4B shows the abundant, uniform staining across the organoid section, indicating ubiquitous mineralization and robust osteogenesis.

### 3.2.2. MSC-Derived Cartilage Organoids

The CM-MSC medium used to generate cartilage organoids contained transforming growth factor beta-3 (TGFβ3), a chondrogenic growth factor widely used to induce MSC chondrogenesis (Table S2) [9]. The control group was cultured in the absence of TGFβ3, but in the presence of kartogenin (Table S2), a small molecule reported to be capable of promoting chondrogenic differentiation of MSCs [34]. RT-qPCR was carried out to evaluate the expression of three key chondrogenic marker genes, including collagen type II (*COL2*), aggrecan (*ACAN*), and SRY-box transcription factor 9 (*SOX9*), in the MSC-derived cartilage organoids. The expression levels of these genes were significantly higher in the cartilage organoids than in the control group (Figure 4C). The RT-qPCR results also revealed that TGFβ3 was more effective in inducing MSC chondrogenesis than KGN in the MSC-ECM system.

Safranin O/Fast Green staining results further confirmed the deposition of sulfated glycosaminoglycans (GAGs), a key ECM component of articular cartilage, in the cartilage organoids (Figure 4D). By contrast, no positive staining could be observed in the control group.

### 3.2.3. MSC-Derived Adipose Organoids

As shown in Figure 4E, the engineered adipose organoids showed high levels of adipogenic gene expression, including peroxisome proliferator-activated receptor gamma variant 1(*PPARG1*), lipoprotein lipase (*LPL*), and *ADIPSIN*. BODIPY staining confirmed the presence of lipid droplets in the adipose organoids (Figure 4F). The stronger staining intensity at the periphery suggests a higher level of adipogenesis near the surface of the 3D organoid.

### 3.3. iMPC-Derived Organoids

### 3.3.1. iMPC-Derived Bone Organoids

Based on the results of MSC-derived bone organoids, HANRs were also incorporated into the iMPC-ECM constructs, which were then cultured in OM-iMPC to induce the formation of bone organoids. The control group had no HANRs and was cultured in GM-iMPC.

The osteogenic differentiation of iMPCs in the bone organoids was indicated by the high levels of osteogenic gene expression, such as *OCN*, *ALP*, and *RUNX2* (Figure 5A). This was further confirmed by Alizarin Red staining, with strong staining intensity showing high matrix mineralization in the bone organoids (Figure 5B).

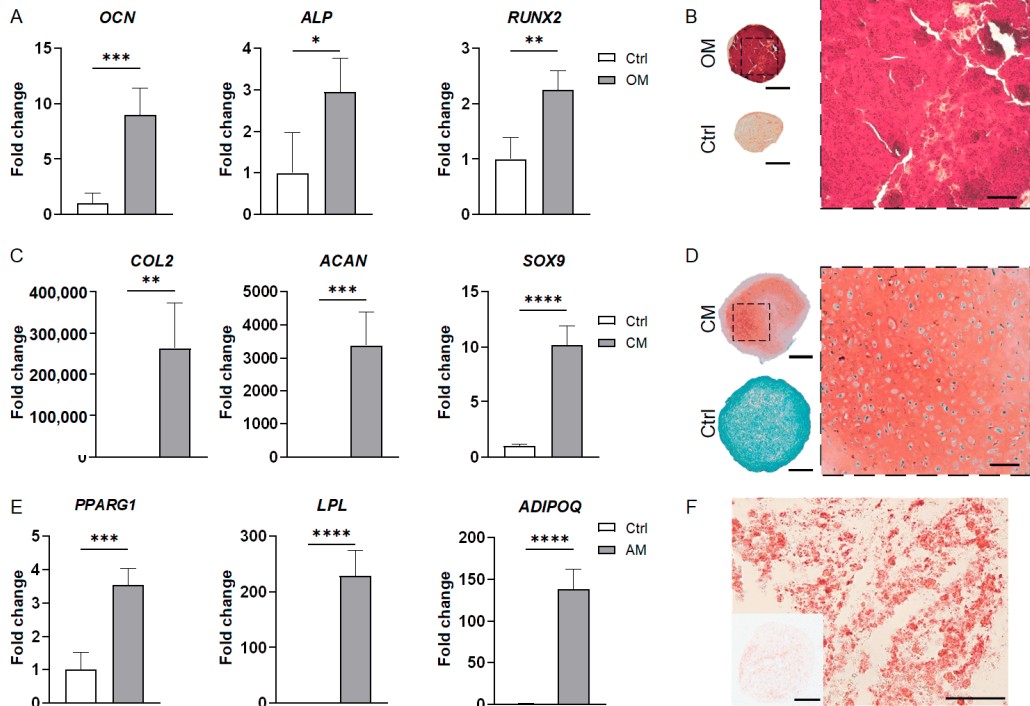

**Figure 5.** Characterization of iMPC-derived bone, cartilage, and adipose organoids. (**A**) Expression levels of osteogenic markers in the bone organoids compared to the control group (set at 1). (**B**) Alizarin Red staining images of the bone organoid and control sample. A magnified view of the dashed line square in the whole-section image is shown on the right. (**C**) RT-qPCR results showing the expression of chondrogenic marker genes in the cartilage organoids compared to the control group (set at 1). (**D**) Safranin O staining images of the cartilage organoid and control group. A magnified view of the dashed line square in the whole-section image is shown on the right. (**E**) Expression of adipogenic marker genes in the adipose organoids and control group, normalized to the control group. (**F**) Oil Red O staining images of the adipose organoids. All RT-qPCR data were analyzed by Student's *t*-test (N = 4 replicates). *, $p < 0.05$; **, $p < 0.01$; ***, $p < 0.001$; ****, $p < 0.0001$. Scale bar = 500 μm for all whole-section images and 100 μm for others.

### 3.3.2. iMPC-Derived Cartilage Organoids

Unlike MSCs, chondrogenesis of iMPCs was reported to require bone morphogenetic protein (BMP) in addition to TGFβ [30,35]. Our previous study found that when utilized together with TGFβ3, BMP6 is a potent inducer of iMPC chondrogenesis [30]. Therefore, BMP6 was supplemented in CM-iMPC in the generation of cartilage organoids (Table S2).

Figure 5C shows high expression of chondrogenic markers, including *COL2*, *ACAN*, and *SOX9* in the cartilage organoids. GAG deposition in the ECM of cartilage organoids was clearly seen in the Safranin O/Fast Green staining images (Figure 5D), signifying robust chondrogenesis by iMPCs.

### 3.3.3. iMPC-Derived Adipose Organoids

The adipose organoids engineered from iMPCs were characterized by RT-qPCR and histology. As shown in Figure 5E, compared with the control group, adipose organoids expressed significantly higher levels of *PPARG1*, *LPL*, and adiponectin (*ADIPOQ*). The Oil Red O staining images revealed the presence of oil droplet deposits within the adipose organoids (Figure 5F).

### 3.3.4. iMPC-Derived Osteochondral Organoids

Figure 6A shows that the osteochondral organoids expressed higher levels of *ALP* and *RUNX2* than the bone organoids (Figure 6A), indicating robust osteogenesis in these biphasic constructs. Compared with the cartilage organoids, the osteochondral organoids expressed similar levels of *COL2* and *SOX9*, but showed lower *ACAN* expression (Figure 6A).

The ECM components of the osteochondral organoids were analyzed by Alizarin Red and Safranin O staining. As shown in Figure 6B,C, these organoids resemble core–shell structures, with a highly mineralized core and a shell rich in GAGs.

Furthermore, IHC images showed that OCN was robustly expressed in the osseous core of the osteochondral organoid, which resembled the high OCN level in the native subchondral tissue (Figure 6D). The chondral shell of the osteochondral organoid showed rich COL2 deposition, akin to the high COL2 levels observed in the superficial layer of the native cartilage (Figure 6E). The osseous core of the osteochondral organoid was negative for COL2 (Figure 6E; images of negative controls provided in Figure S1).

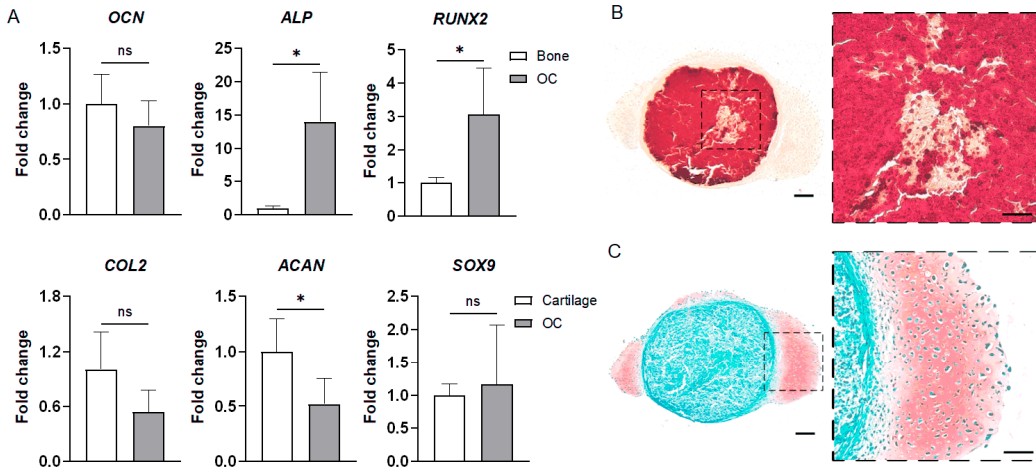

**Figure 6.** *Cont*.

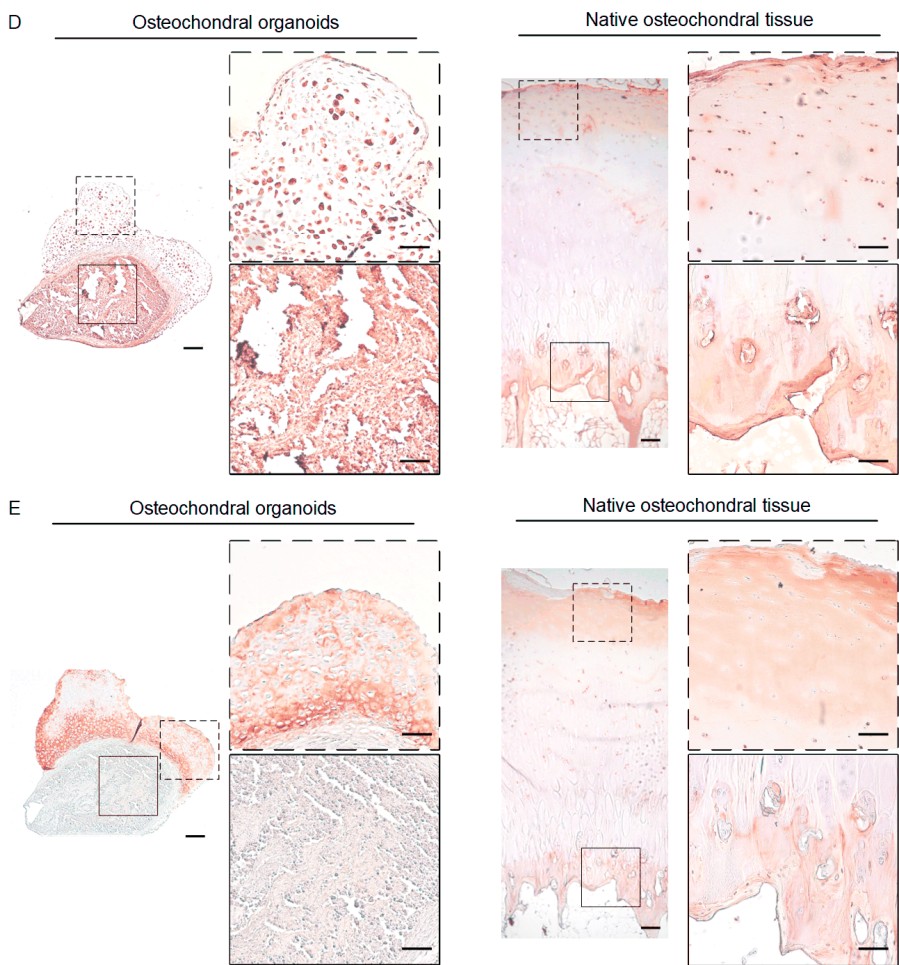

**Figure 6.** Characterization of iMPC-derived osteochondral organoids. (**A**) The expression levels of osteogenic and chondrogenic genes in the osteochondral organoids relative to those in the bone and cartilage organoids, respectively. The values of the osteochondral organoids were normalized to those of bone (for osteogenic markers) and cartilage (for chondrogenic markers) organoids to obtain the fold change values. Statistical analysis was conducted by Student's $t$-test. N = 4 replicates. *, $p < 0.05$. (**B**) Alizarin Red staining revealed a highly mineralized core in the osteochondral organoid. (**C**) Safranin O/Fast Green staining demonstrates the formation of a cartilage layer on the surface. (**D**) IHC staining of OCN in the osteochondral organoid and native osteochondral tissue. (**E**) IHC staining of COL2 in the osteochondral organoids and native osteochondral tissue. The images with dashed/solid line border in (**B**–**E**) are magnified views of the corresponding dashed/solid line squares in the low-magnification images on the left. Scale bars in (**B**–**E**) = 200 μm (low-magnification images) and 100 μm (magnified views).

## 4. Discussion

The lack of effective models to study joint disease mechanisms and develop disease-modifying drugs is a principal cause of the significant unmet medical need for treating these highly prevalent disorders. In this study, the organoid technology, an emerging field being utilized for understanding tissue/organ development, biology, and etiologies, was employed to generate models of tissues of the articular joint using human MSCs and iPSCs. In addition, a novel approach was proposed to engineer osteochondral organoids from iMPCs. This study verified the feasibility of producing articular tissue-mimicking organoids via the self-aggregation and subsequent differentiation of human stem cells within their own ECM. The organoids' resemblance to the corresponding native tissues was confirmed by gene expression analysis, histology, and IHC. The findings of this study

inform the potential application of articular tissue-mimicking organoids in joint repair and regeneration, disease modeling, and drug development.

The current study differs from previous investigations into organoids derived from cell-ECM constructs in several significant ways [23]. First, the iMPC-ECM system was utilized for the first time to engineer organoids that mimic articular tissues. Thus, with the use of patient-specific iPSCs, personalized organoids can be obtained as an in vitro platform for the development of precision medicine. Second, this study verified the feasibility of generating adipose organoids from stem cell-ECM constructs. Finally, osteochondral organoids with a "bone core" and "cartilage shell" were produced for the first time, and HANRs were leveraged to enhance osteogenesis quality in the bone and osteochondral organoids.

HANRs as osteoinductive nanoparticles were found to be more effective in inducing MSC osteogenesis than the commonly used osteogenic medium. With the capability of driving stem cell lineage specification, such bioactive nanomaterials would be particularly useful when utilized in different types of organoids that are fluidically coupled in vitro, for example, as interconnected organoids-on-chips. A key challenge to establishing these microphysiological systems is the maintenance of mature tissue phenotypes, as a circulating shared medium is usually necessary. The incorporation of functional nanomaterials can potentially contribute to the individual phenotypic stability of the interconnected organoids, thus minimizing or even obviating the use of tissue-specific media in the microphysiological systems.

Varying staining intensity was observed across the Safranin O-stained cartilage organoid section (Figure 4D), indicating non-uniform GAG deposition. A similar phenomenon was seen at the periphery of the iMPC-derived OC organoids (Figure 6C). These observations possibly resulted from limited nutrient diffusion to the bottom of the organoids as they were statically cultured. Such inhomogeneity in ECM composition can potentially be eradicated by using dynamic culture platforms such as rotating wall vessels. Dynamic culture may also lead to more robust lipid droplet deposition at the center of the MSC-derived adipose organoids, which showed a lower level of adipogenesis than the periphery (Figure 4F), possibly caused by insufficient supply of adipogenic factors to the cells near the center.

Using a two-stage induction process, osteochondral organoids were successfully generated from HANR-incorporated iMPC-ECM constructs. Our findings suggest that after 21 days of osteogenic induction, the cells or a subpopulation of them did not commit to an osteoblastic lineage, but rather retained their chondrogenic differentiation potential. This phenomenon is consistent with our previous report of the transdifferentiation potential of MSCs [36], and the identification of candidate genes involved in regulating MSC differentiation multipotency [37]. An in-depth investigation into the temporal changes in cell behaviors would be highly valuable to understand the biology underlying osteochondral organoid formation. In addition, the induction protocol for engineering osteochondral organoids can possibly be optimized, for example, by tuning the durations of the two stages.

The creation of osteochondral organoids from stem cells has also been explored in previous studies. Hall and colleagues engineered a patterned construct with a cartilaginous layer and a "callus organoid" layer [24]. However, their bottom-up approach required the predifferentiation of different cell sources (iPSCs and periosteum-derived cells) followed by assembly. Guilak's group used murine iPSCs to produce osteochondral organoids by sequentially exposing the cells to TGFβ3 and BMP2 to mimic the endochondral ossification process [38]. The resultant osteochondral organoids possessed a cartilaginous region at the center and a calcified component as the shell. As articular cartilage caps the subchondral bone in the native joint, osteochondral organoids with a cartilaginous layer surrounding the bony region, as reported here, better mimic native anatomy and represent a desirable feature for in vitro osteochondral models.

Extended research in a number of interesting areas is worth pursuing in the future. First, modeling of joint diseases can be conducted by, for example, inducing cartilage organoid degeneration by exposure to biochemical or biomechanical insults, offering a

novel in vitro platform for studying joint pathophysiology. Second, the spatial variation in cell differentiation and maturation (as revealed by histological analyses) can be examined by using techniques such as single-cell sequencing and spatial transcriptomics. Third, the osteochondral organoids may be used as more physiologically relevant 3D cell constructs to examine the potential involvement of genes implicated in the regulation of MSC multipotency [37]. Finally, as OA and RA are both whole-joint diseases, it would be paramount to establish a multi-organoid platform with high physiological and clinical relevance for in-depth, mechanistic studies and preclinical evaluation of potential DMOADs and DMARDs. A possible approach to connecting these organoids and enabling their reciprocal communication is by incorporating them in organoid-on-a-chip systems [9].

## 5. Conclusions

This study reported the generation and characterization of articular tissue-mimicking organoids from human MSCs and iMPCs. The stem cell-secreted ECM provided a 3D natural and biocompatible microenvironment that supported cell differentiation to generate bone, cartilage, adipose, and osteochondral organoids. Unlike previous studies, iPSC-derived osteochondral organoids with a bony core and a cartilaginous shell were successfully generated by leveraging osteoinductive bioceramic nanoparticles. The in vitro engineered articular tissue-mimicking constructs hold promise as a versatile platform for mechanistic studies on joint biology and pathology as well as translational investigations into disease-modifying drugs and precision medicine.

**Supplementary Materials:** The following supporting information can be downloaded at: https://www.mdpi.com/article/10.3390/organoids1020011/s1, Table S1: Composition of media used for cell maintenance and tri-lineage differentiation assay; Table S2: Composition of media used to induce MSC and iMPC differentiation to form different organoids; Table S3: Primer sequences used for RT-qPCR. Figure S1: Negative controls for IHC, using mouse immunoglobulin G (IgG) isotypes (0.5 μg/mL) in place of the primary antibodies. (A) iMPC-derived osteochondral organoid. (B) Native osteochondral tissue. Scale bar = 200 μm.

**Author Contributions:** Conceptualization, Z.A.L., H.L. and R.S.T.; methodology, Z.A.L., J.S., S.X., E.N.L., H.Y. and K.R.; software, Z.A.L.; validation, Z.A.L.; formal analysis, Z.A.L., H.L. and R.S.T.; investigation, Z.A.L.; resources, H.L.; data curation, Z.A.L.; writing—original draft preparation, Z.A.L.; writing—review and editing, H.L. and R.S.T.; visualization, Z.A.L.; supervision, H.L. and R.S.T.; project administration, H.L. and R.S.T.; funding acquisition, H.L. and R.S.T. All authors have read and agreed to the published version of the manuscript.

**Funding:** This research was funded by the Department of Orthopaedic Surgery at the University of Pittsburgh School of Medicine (to H.L.). R.S.T. is supported by the Lee Quo Wei and Lee Yik Hoi Lun Professorship in Tissue Engineering and Regenerative Medicine of The Chinese University of Hong Kong and the Centre for Neuromusculoskeletal Restorative Medicine of the InnoHK Cluster (Innovation and Technology Commission, Hong Kong SAR, China).

**Institutional Review Board Statement:** Human bone marrow-derived MSCs were isolated from de-identified donors according to an exempted protocol approved by the Institutional Review Board (IRB), University of Washington & University of Pittsburgh (approval code PRO17070639).

**Informed Consent Statement:** Not applicable.

**Data Availability Statement:** The data supporting the reported results are available from the corresponding author upon reasonable request.

**Acknowledgments:** The authors thank Paul Manner (University of Washington) for providing the clinical specimens and Jian Tan (University of Pittsburgh) for her assistance with MSC isolation and maintenance.

**Conflicts of Interest:** The authors declare no conflict of interest. The funders had no role in the design of the study; in the collection, analyses, or interpretation of data; in the writing of the manuscript, or in the decision to publish the results.

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
