# Peer review of "Articular Tissue-Mimicking Organoids Derived from Mesenchymal Stem Cells and Induced Pluripotent Stem Cells"

_2674-1172, doi:10.3390/organoids1020011_

Round 1

Reviewer 1 Report

Dear authors, 
I congratulate you for the article, which is interesting from a scientific point of view and well written.

The materials and methods are appropriate, the results are clearly presented.

I would like however to see in the discussion section a better analysis of thee results compared to relevant scientific literature 

Author Response

I congratulate you for the article, which is interesting from a scientific point of view and well written.

The materials and methods are appropriate, the results are clearly presented.

I would like however to see in the discussion section a better analysis of thee results compared to relevant scientific literature

Response: We thank the Reviewer for the positive feedback on our manuscript. We have now further analyzed the results and referenced relevant scientific literature in the Discussion section. Please see Line 405-415.

Reviewer 2 Report

The manuscript entitled “Articular Tissue-Mimicking Organoids Derived from Mesenchymal Stem Cells and Induced Pluripotent Stem Cells” by Zhong Alan Li and colleagues describes different protocols for the generation of adipose tissue, cartilage as well as bone organoids. Moreover, the authors present a novel protocol for the generation of osteochondral organoids as 3D tissue models to study joint disorders. Primary bone derived MSCs as well as iPSCs were used as cell source and hydroxyapatite nanorods were incorporated to enhance tissue mineralization.

The manuscript is interesting, well written and sound. However, I have some comments and suggestions:

1.)  The authors state in the legend of figures 3-5 that 4 technical replicates were performed. I suggest doing at least 3 biological replicates to get insights into the reproducibility of organoid formation.

2.)  In general, reproducibility and robustness of protocols for organoid formation is an important issue. The authors should provide more data and quantification regarding that.

3.)  How many different iPSC lines or MSC preparations were used to conduct the experiments. Is there any line-to-line variation in the differentiation outcome?

4.)  For me, the osteochondral organoids are the most interesting part of the manuscript. It would be great if the authors would provide more histological data and compare their organoids to the osteochondral interface in vivo. It consists of articular cartilage, calcified cartilage and subchondral bone forming an integrated functional unit. The cartilage consists of three zones, the superficial zone, the transition zone, and the deep zone, followed by the tidemark and the adjoining calcified cartilage. The subchondral bone layer lies at the bottom and includes the subchondral bone plate. It would be interesting to know, if a similar layering can be also observed in the organoids.

5.)  In the methods section, the protocol could be described in more detail. The authors often refer to already published protocols, e.g., in Biomaterials. As the presented manuscript is a methods paper, I would include the detailed protocol into the manuscript.

6.)  Minor comments:

a.    In Fig 2C, different magnifications are used in the upper and lower pictures (MSCs vs. iMPCs). It would be better to use the same magnification.

b.    In Fig.3 and Fig.4 different adipocyte markers are used for Q-PCR analyses (ADIPISIN in Fig.3 and ADIPOQ in Fig.4). Is there any specific reason for that?

c.    I would include Figure S1 into the main manuscript.

d.    Primer pairs used in Figure S1 are not listed in Table S3

Author Response

The manuscript entitled “Articular Tissue-Mimicking Organoids Derived from Mesenchymal Stem Cells and Induced Pluripotent Stem Cells” by Zhong Alan Li and colleagues describes different protocols for the generation of adipose tissue, cartilage as well as bone organoids. Moreover, the authors present a novel protocol for the generation of osteochondral organoids as 3D tissue models to study joint disorders. Primary bone derived MSCs as well as iPSCs were used as cell source and hydroxyapatite nanorods were incorporated to enhance tissue mineralization.

The manuscript is interesting, well written and sound. However, I have some comments and suggestions:

 Response: We thank the Reviewer for the careful review of our manuscript and appreciate the very helpful comments and suggestions.

1.)  The authors state in the legend of figures 3-5 that 4 technical replicates were performed. I suggest doing at least 3 biological replicates to get insights into the reproducibility of organoid formation.

Response: We would like to explain what we meant by technical replicates. The four replicates used in our study were four distinct organoids, engineered from the same cell sources. We also repeated the experiments to confirm the trends observed.

Our definition of “technical replicates” was based on the article by Blainey et al. (Nat Methods 2014, 11, 879–880, https://www.nature.com/articles/nmeth.3091). As we pooled MSCs isolated from 20 donors (see Section 2.1) and used a well-established iPSC line (see Section 2.2), we treated the 4 distinct organoid samples we produced as “technical replicates”. However, we are aware that in many studies, such samples were referred to as “biological replicates”.

While we do not think “biological replicate” is the appropriate term to use, the experiment design supports the reproducibility of the data. To avoid confusion, we have changed “technical replicates” to “replicates” and highlighted the changes in the revised manuscript.

2.)  In general, reproducibility and robustness of protocols for organoid formation is an important issue. The authors should provide more data and quantification regarding that.

Response:  We agree with the Reviewer that reproducibility and robustness of the protocols are critical. As we mentioned in our Response to the last point, the replicates we used in this study represent distinct samples cultured independently; we also repeated our experiments to confirm the observed trends. Furthermore, we have carried out additional immunostaining of COL2 and OCN to further analyze the osteochondral organoids and compared them with native osteochondral tissues (Figure 6 D, E in the revised manuscript).

3.)  How many different iPSC lines or MSC preparations were used to conduct the experiments. Is there any line-to-line variation in the differentiation outcome?

Response: For MSCs, we created a cell pool with cells from 20 different donors. The pooled cells were used to derive the organoids. As reported in our previous study, MSCs from all donors have tri-lineage differentiation potential in both 2D and 3D cultures, and there is donor-to-donor variability in terms of the cells’ differentiation potential  (Adv. Sci. 2022, 9(21), 2270133). However, the focus of the current study is the feasibility of producing articular tissue-mimicking organoids using the reported protocols, and donor-dependent differentiation outcome can be the topic of a separate investigation. For iPSCs, we used a well-established iPSC line, which is well characterized and has been used in several previous studies in our lab (e.g., Stem Cells Dev. 2014, 23, 1594-1610; Front. Bioeng. Biotechnol. 2019, 7, 411).

4.)  For me, the osteochondral organoids are the most interesting part of the manuscript. It would be great if the authors would provide more histological data and compare their organoids to the osteochondral interface in vivo. It consists of articular cartilage, calcified cartilage and subchondral bone forming an integrated functional unit. The cartilage consists of three zones, the superficial zone, the transition zone, and the deep zone, followed by the tidemark and the adjoining calcified cartilage. The subchondral bone layer lies at the bottom and includes the subchondral bone plate. It would be interesting to know, if a similar layering can be also observed in the organoids.

Response: We thank the Reviewer for this suggestion. We have further analyzed the osteochondral organoids and compared them with native osteochondral tissues using immunohistochemistry, and the new data are presented in Figure 6D, E and discussed in Lines 331-336.

5.)  In the methods section, the protocol could be described in more detail. The authors often refer to already published protocols, e.g., in Biomaterials. As the presented manuscript is a methods paper, I would include the detailed protocol into the manuscript.

Response: We have now added more details on iPSC production, HANR synthesis, and IHC in the revised manuscript and highlighted the changes (Lines 99-100, 138-141, 175-194).

6.)  Minor comments:

  1. In Fig 2C, different magnifications are used in the upper and lower pictures (MSCs vs. iMPCs). It would be better to use the same magnification.

Response: We thank the Reviewer for the suggestion. We have now used the same magnification for all images in Fig. 2C (Lines 210-215).

  1. In Fig.3 and Fig.4 different adipocyte markers are used for Q-PCR analyses (ADIPISIN in Fig.3 and ADIPOQ in Fig.4). Is there any specific reason for that?

Response: As both markers are highly relevant to the adipogenic differentiation of stem cells, we believe the three gene markers we used for each group collectively support the adipogenesis of the cells. While the same markers should ideally be used for both groups, comparison of the relative expression levels between the two cell types was not the intent.

  1. I would include Figure S1 into the main manuscript.

Response: We have moved Figure S1 to the main manuscript as Figure 3.

  1. Primer pairs used in Figure S1 are not listed in Table S3

Response: We have added the missing primer sequences for genes in Figure S1 (Figure 3 in the revised manuscript), including BMP2, OPN, and BSP2. Please refer to updated Table S3 in the SI.

Reviewer 3 Report

The authors clearly prsent that osteochindral or adipose 3D structures can be generated.

Organoids are sparsely characterized, admittidly this is one of the least complicated differentiation paths to go with MSCs or iMSCs. The results are well depicted.

Overall, the study lacks a proper comparison between the to cell types and their progeny. It would have been interesting to see differences in organoids derived from multi- or pluripotency.

The word organoids is a bit misleading as the authors simply took their cultures in a floating surrounding.

The study is not novel as all results have been shown by others already. Other studies are already at the stage of vascularizing the respective organoids etc.

Author Response

The authors clearly prsent that osteochindral or adipose 3D structures can be generated.

Response: We thank the Reviewer for the positive feedback on the clear presentation and depiction of the organoids derived from two types of cells.

Organoids are sparsely characterized, admittidly this is one of the least complicated differentiation paths to go with MSCs or iMSCs. The results are well depicted.

As the Reviewer pointed out, the approaches employed by us were not complicated, but were effective in engineering MSC- and iMPC-derived organoids. We believe the complexity of the protocols can be increased, for example, to obtain more homogeneous cartilage and adipose organoids in the future.

Overall, the study lacks a proper comparison between the to cell types and their progeny. It would have been interesting to see differences in organoids derived from multi- or pluripotency.

The focus of this study is on: (1) the development of appropriate protocols for organoid production, especially for engineering the iMPC-derived organoids, as few previous studies looked into generating the reported organoid types using the iMPC-ECM system; and (2) the characterization of the phenotypes in the obtained organoids. To address the Reviewer’s concerns on organoid characterization, we carried out additional immunostaining of COL2 and OCN to further confirm the phenotypes of the osteochondral organoids (Figure 6D, E). We believe further comparative evaluation of the two types of organoids (by, for example, spatial transcriptomics) would be interesting, but is beyond the scope of this study.

The word organoids is a bit misleading as the authors simply took their cultures in a floating surrounding.

The definition of organoid has been evolving, and the term has been used to refer generally to 3D multicellular structures recapitulating functions of native tissues/organs. For example, in several recent studies, 3D constructs generated using similar techniques were referred to as organoids (e.g., Biomaterials 2021, 273, 120820; Adv. Sci. 2020, 7, 1902295). Therefore, we believe that the constructs we studied can be called organoids in a broad sense.

The study is not novel as all results have been shown by others already. Other studies are already at the stage of vascularizing the respective organoids etc.

As we mentioned in the Discussion section, this study differs from previous investigations in several significant ways. First, the iMPC-ECM system was utilized for the first time to engineer organoids that mimic articular tissues. Second, this study verified the feasibility of generating adipose organoids from stem cell-ECM constructs. Finally, osteochondral organoids with a “bone core” and “cartilage shell” were produced for the first time, and HANRs were leveraged to enhance the quality of osteogenesis in the bone and osteochondral organoids.

While vascularization is a key consideration in engineering bone organoids, as shown in a recent study by our group (Biomaterials 2022, 283: 121451), vasculature may not be incorporated in organoids emulating healthy articular cartilage, which is naturally avascular.